# Slow and Steady, or Hard and Fast? A Systematic Review and Meta-Analysis of Studies Comparing Body Composition Changes between Interval Training and Moderate Intensity Continuous Training

**DOI:** 10.3390/sports9110155

**Published:** 2021-11-18

**Authors:** James Steele, Daniel Plotkin, Derrick Van Every, Avery Rosa, Hugo Zambrano, Benjiman Mendelovits, Mariella Carrasquillo-Mercado, Jozo Grgic, Brad J. Schoenfeld

**Affiliations:** 1School of Sport, Health and Social Sciences, Solent University, Southampton SO14 0YN, UK; james.steele@solent.ac.uk; 2Health Sciences Department, CUNY Lehman College, Bronx, NY 10468, USA; danielplotkin96@gmail.com (D.P.); vaneverd@uwindsor.ca (D.V.E.); averyrosa1@gmail.com (A.R.); hugo9523@gmail.com (H.Z.); benjimanbm1@gmail.com (B.M.); mariellacm123@gmail.com (M.C.-M.); 3Institute for Health and Sport, Victoria University, Melbourne, VIC 8001, Australia; jozo.grgic@live.vu.edu.au

**Keywords:** intensity of effort, fat loss, fat mass, body fat, lean mass

## Abstract

Purpose: To conduct a systematic review and multilevel meta-analysis of the current literature as to the effects of interval training (IT) vs moderate intensity continuous training (MICT) on measures of body composition, both on a whole-body and regional level. Methods: We searched English-language papers on PubMed/MEDLINE, Scopus, CINAHL, and sportrxiv for the following inclusion criteria: (a) randomized controlled trials that directly compared IT vs MICT body composition using a validated measure in healthy children and adults; (b) training was carried out a minimum of once per week for at least four weeks; (c) published in a peer-reviewed English language journal or on a pre-print server. Results: The main model for fat mass effects revealed a trivial standardized point estimate with high precision for the interval estimate, with moderate heterogeneity (−0.016 (95%CI −0.07 to 0.04); *I*^2^ = 36%). The main model for fat-free mass (FFM) effects revealed a trivial standardized point estimate with high precision for the interval estimate, with negligible heterogeneity (−0.0004 (95%CI −0.05 to 0.05); *I*^2^ = 16%). The GRADE summary of findings suggested high certainty for both main model effects. Conclusions: Our findings provide compelling evidence that the pattern of intensity of effort and volume during endurance exercise (i.e., IT vs MICT) has minimal influence on longitudinal changes in fat mass and FFM, which are likely to minimal anyway. Trial registration number: This study was preregistered on the Open Science Framework.

## 1. Introduction

The relative components of fat mass and fat-free mass in the body, collectively termed body composition, has important implications for human health. Excessive levels of body fat show a high correlation with a panoply of disease states, including cardiovascular diseases, metabolic disorders, certain cancers, osteoarthritis, and respiratory conditions [1]. Alternatively, low levels of fat-free mass are associated with a loss of strength, functional capacity, and reduced bone mineral density [2,3,4], impairing both the quality and quantity of life [1]. There is an interaction between these two components, whereby the combination of low levels of fat-free mass (FFM) and high levels of body fat potentiate each other, maximizing their impact on disability, morbidity, and mortality [5].

Exercise is commonly recommended as an intervention to improve body composition [6,7]. Interventional strategies often employed for this purpose include the following patterns.

Moderate intensity continuous training (MICT), herein operationally defined as moderate intensity of effort exercise (<80% peak heart rate or aerobic capacity) performed over a longer (relative to interval training bouts) single bout.

Interval training (IT), herein operationally defined as exercise performed in multiple shorter (relative to continuous training) bouts interspersed with recovery periods either at lower intensities of effort, or as complete rest.

IT is often subclassified into high intensity interval training (HIIT), herein operationally defined as high intensity of effort exercise (approximately >80% peak heart rate or aerobic capacity) performed in multiple shorter bouts interspersed with recovery periods either at lower intensities of effort or as complete rest, and sprint interval training (SIT), herein operationally defined as maximal intensity of effort exercise (‘all out’ sprint) performed in multiple shorter bouts interspersed with recovery periods either at lower intensities of effort or as complete rest.

Although both MICT and IT show efficacy in improving body composition, controversy exists as to whether one strategy is superior to the other for this purpose. For example, an earlier meta-analysis by Keating et al. [8] reported little difference between MICT and IT for body fat reduction, highlighting that, over the short term, neither intervention produced clinically meaningful changes. Following this, Viana et al. [9] conducted a meta-analysis with results showing that IT produced a 28.5% greater reduction in fat mass than MICT. However, the paper was criticized for various methodological issues [10], ultimately leading to its retraction. More recently, Sultana et al. [11] carried out a meta-analysis that included a comparison of IT vs MICT. The analysis did not find a benefit to low-volume IT on measures of body composition when compared with MICT. However, they limited their analysis to only single measures per study of the constructs of interest (i.e., total body fat mass, body fat percentage, and lean body mass), whereas many studies often report several measures (e.g., regional measures). Furthermore, although several studies have also compared the effects of IT and MICT in younger populations, they limited the analysis to adults. Additionally, it is not clear from their analysis which pre-post test correlations were imputed and used for effect size calculations. The magnitude of pre-post test correlations used in calculations of pre-post control group design effect sizes using pooled baseline standard deviations can impact the heterogeneity determined in the meta-analysis [12]. Thus, although the standardized point estimates of Sultana et al. [11] models generally suggested little difference between conditions, the accompanying interval estimates for most outcomes included small effects in favor of either IT or MICT. Furthermore, their models had essentially no heterogeneity, although this may be the result of imputation of pre-post correlations that were relatively low. Application of multilevel meta-analytic models with robust variance estimation to handle multiple effects per study might yield a greater precision of estimates [13], and thus help to confirm whether small differences do in fact exist, and if so, in which direction. Additionally, extraction of information to permit calculation of pre-post test correlations within groups (i.e., see Higgins et al. [14]) would allow for a better estimate of the population pre-post test correlations and may reveal heterogeneity not identified in previous analyses. Lastly, although Sultana et al. [11] explored ‘within-condition’ effects for IT in studies that included a non-exercising control condition, they did not similarly explore this outcome for MICT training.

It also has been speculated that specific exercise-induced effects might occur for hypertrophy and regional fat mass. Endurance exercise may have beneficial effects on muscle hypertrophy, similar to that of resistance training [15], and some researchers highlight that IT, in particular, may produce a potent anabolic stimulus [16]. Furthermore, it has been suggested that IT may be more effective than MICT for abdominal fat mass reduction [17]. However, to our knowledge, no previous review has pooled data from research that directly compares changes in FFM between IT and MICT, nor specifically examined regional effects on changes in fat mass. 

Lastly, although prior meta-analyses have considered between-conditions comparison of mean intervention effects [11], whether or not differences in the variance of treatment responses are present has been relatively less explored. A recent meta-analysis of aerobic exercise in overweight and obese children and adolescents found no evidence of ‘true’ inter-individual response variation in fat loss [18]. However, numerous studies have purported that there may be inter-individual response variation to IT and MICT for a range of outcomes [19,20,21], and indeed it has been argued that such variation may mask differences between IT and MICT for fat loss [22]. Thus, we also sought to examine whether there is evidence of ‘true’ inter-individual response variation for body composition outcomes for both IT and MICT [23,24].

Given the gaps in the current literature, the purpose of this paper was to conduct a systematic review and multilevel meta-analysis of the current literature as to the effects of IT vs MICT on measures of body composition, both on a whole-body and regional level. Secondarily, we sought to determine if intensity of effort influences exercise adherence and/or adverse events, as well as whether inter-individual response to IT and MICT influences changes in body composition.

## 2. Material and Methods

This systematic review was conducted in accordance with the guidelines of the “Preferred Reporting Items for Systematic Reviews and Meta-Analyses” (PRISMA) [25]. The study was preregistered on the Open Science Framework (https://osf.io/dq784), where the detailed prespecified methodological protocol can be viewed.

### 2.1. Inclusion/Exclusion Criteria

We included studies that met the following criteria: (a) randomized controlled trials (both within- and between-group designs) that directly compared IT vs MICT (both with and without adjuvant dietary interventions) for body composition using a validated measure (DXA, BodPod, hydrostatic weighing, BIA, skinfolds, ultrasound, magnetic resonance imaging, and computerized tomography) in healthy children and adults; (b) training was carried out a minimum of once per week for at least four weeks; (c) published in a peer-reviewed English language journal or on a pre-print server. We excluded studies that employed: (a) participants with co-morbidities that might impair aerobic capacity (respiratory conditions, musculoskeletal injury); and (b) an unbalanced resistance training component (e.g., one group performs resistance training whereas the other does not). Note, our original pre-registration failed to specify the particular intensity of effort and operationalization of this variable for determination of whether an IT intervention could be considered ‘HIIT’. However, a small number of studies identified employed intensities of >75% of peak heart rate or aerobic capacity for their IT conditions. Given our omission of specificity in pre-registration, we felt that these studies should be included, as there was still a reasonable difference in intensity of effort compared with the MICT conditions (typically <60%).

### 2.2. Search Strategy

We carried out a comprehensive search of the PubMed/MEDLINE, Scopus, CINAHL, and sportrxiv databases using the following Boolean string: (interval training OR intermittent training OR high intensity OR sprint interval training OR aerobic interval training OR HIIT OR HIIE OR high intensity interval training OR high-intensity interval training OR high intensity interval exercise OR high intensity intermittent exercise OR high-intensity intermittent exercise OR high intensity intermittent training OR high-intensity intermittent training) AND (continuous training OR moderate-intensity continuous exercise OR moderate intensity continuous exercise OR moderate-intensity continuous training OR moderate intensity continuous training OR endurance training) AND (body fat OR adiposity OR body composition OR abdominal fat OR visceral fat OR adipose tissue OR fat mass OR fat-free mass OR lean body mass OR lean mass OR muscle mass). Moreover, we screened the reference lists of articles retrieved to uncover any additional studies that might meet inclusion criteria, as described by Greenhalgh and Peacock [26]. The search was finalized on 6 March 2021; Figure 1 illustrates a flow chart of the search process.

### 2.3. Screening/Coding of Studies

Search/screening was carried out separately by two researchers (DP and AR). These researchers read all titles and abstracts and then reviewed full texts for papers deemed relevant based on their title and abstract. Decisions then were made as to whether a study warranted inclusion based on the stated criteria. Any disputes on the inclusion of a given study were settled by a third researcher (MCM).

After determining which studies met inclusion, two researchers (DV and HZ) separately coded the following variables for each study: authors, title and year of publication, sample size, sex, body mass index (BMI), training status, age, description of the training intervention (duration, intensity, frequency, modality), work matched (yes/no), nutrition controlled (yes/no), method for body comp assessment (e.g., DXA, BodPod, BIA, hydrostatic weighing, skinfolds, MRI, CT, ultrasound), number of adverse effects associated with the training intervention, adherence to the given training program, mean pre- and post-study body composition values in addition to pre-post change scores with the corresponding standard deviation or standard error, and where change score standard deviations were not reported we extracted information to allow their calculation, including confidence intervals for change scores or within-group pre-post t statistics or *p* values (where *p* values were reported only to the studies’ level of alpha (e.g., *p* < 0.05) we took this as a conservative value). In cases where body composition data were not reported numerically, we either extracted the data from graphs when available via online software, or attempted to contact the study’s authors. Coding was cross-checked between reviewers, with any discrepancies resolved by mutual consensus. Consistent with the guidelines of Cooper et al. [27], 30% of the included studies were randomly selected for re-coding to assess for potential coder drift by a third researcher (BM). Agreement was calculated by dividing the number of variables coded the same by the researchers by the total number of variables; acceptance required a mean agreement of 0.90 to avoid re-extraction entirely, and after this was met, only those with differing codes were checked and updated. Extracted data was also double-checked after this process by the lead author (JS) prior to analysis.

### 2.4. Methodological Quality and Certainty of Evidence

Two of the authors independently evaluated each study (JG and BJS) using the 11-point Physiotherapy Evidence Database (PEDro) scale, which has been validated to assess the methodologic quality of randomized trials [28] with acceptable inter-rater reliability [29]. Any discrepancies in agreement on a given scale item were settled by mutual agreement between the researchers. Given that it is infeasible to blind participants and investigators in supervised exercise interventions, we opted to remove assessment items specific to blinding (numbers 5, 6, and 7 in the scale). After eliminating these items, this created a modified 8-point PEDro scale with a maximum value of 7 (the first item is excluded from the total score). The qualitative methodological ratings were amended, similar to those used in previous exercise-related systematic reviews [30], as follows: “excellent” (6–7 points); “good” (5 points); “moderate” (4 points); and “poor” (0–3 points). We also followed the Grading of Recommendations, Assessment, Development and Evaluations (GRADE) framework [31] for evaluating the certainty of evidence with respect to our primary pre-registered outcomes (absolute fat mass, and absolute lean/fat free mass). We used the GRADEpro online tool [32] for this assessment and generation of the summary of findings table. It should be noted though that we did not pre-register the use of the GRADE approach to evaluate the evidence presented but decided a posteriori that the assessment would enhance the ability to draw practical inferences from the data.

### 2.5. Statistical Analyses

Quantitative synthesis of data was performed with the ‘metafor’ [33] package in R (v 4.0.2; R Core Team, https://www.r-project.org/). All analysis code and data are openly available in the Appendix A (https://osf.io/6karz/). Studies were grouped by design (i.e., within- or between-group), and depending on reporting in individual studies, either post or delta comparisons, or pre-post comparison designs [12] for the purposes of appropriate calculation of standardized effects (Hedges’ g) using the escalc function in metafor were carried out. We used the pooled group baseline standard deviation as the numerator as per Morris (29). Standardized effect sizes were interpreted as per Cohen’s [34] thresholds: trivial (<0.2), small (0.2 to <0.5), moderate (0.5 to <0.8), and large (≥0.8). Standardized effects were calculated in such a manner that a positive effect size value favored the IT conditions.

As there was a nested structure to the effect sizes calculated from the studies included (i.e., multiple effects nested within groups and nested within studies), multilevel mixed effect meta-analyses with both study and intra-study groups included as random effects in the model were performed. Cluster robust point estimates and the precision of those estimates using 95% compatibility (confidence) intervals (CIs) were produced, weighted by the inverse sampling variance to account for the within- and between-study variance (τ^2^). Restricted maximal likelihood estimation was used in all models. Two main models were produced for both pre-registered main outcomes (absolute fat mass and FFM), including all standardized effect sizes, to provide a general estimate of the comparative treatment effects. All other models were considered secondary and exploratory analyses.

For all models, we avoided dichotomizing the existence of an effect for the main results and therefore did not employ traditional null hypothesis significance testing, which has been extensively critiqued [35,36]. Instead, we considered the implications of all results compatible with these data, from the lower limit to the upper limit of the interval estimates, with the greatest interpretive emphasis placed on the point estimate. Given the large number of included studies and effects, the main models are visualized here using ordered caterpillar plots to aid interpretation, as opposed to traditional forest plots containing study characteristics. Note that all study characteristics are available in the data file in the Appendix A (https://osf.io/dumq8/), as are more traditional forest plots for the main models (see folder “Outputs and Figures” at https://osf.io/6karz/).

The risk of small study bias was examined visually through contour-enhanced funnel plots. Q and *I*^2^ statistics were also produced and reported [37]. A significant Q statistic is typically considered indicative of effects likely not being drawn from a common population. *I*^2^ values indicate the relative degree of heterogeneity in the effects that are not due to sampling variance and are qualitatively interpreted as: 0–40%: not important, 30–60%: moderate heterogeneity, 50–90%: substantial heterogeneity, and 75–100%: considerable heterogeneity [38]. For within-participant effects, pre-post correlations for measures are often not reported in original studies; thus, for those studies where we had standard deviations for pre-, post-, and change scores (or were able to calculate the latter from confidence intervals, t statistics, or *p* values) we calculated the pre-post correlations directly as:$$r_{pre-post}=\frac{SD_{pre}^{2}+SD_{post\;}^{2}+SD_{change}^{2}}{2\times SD_{pre}\times SD_{post}}$$
and imputed the median correlation coefficient to studies as an appropriate estimate of the population parameter.

In addition to the main models, we secondarily produced models for relative fat and FFM (i.e., as a percentage of body mass), and refit all models using delta scores (i.e., changes) of outcomes in the raw units of measurement (i.e., kilograms and percentages) to facilitate interpretation in a complementary fashion. We also produced models where studies included a non-training control arm that examined the between-condition treatment effects for both IT vs CON, and MICT vs CON, to determine the ‘within-condition’ effect estimates on both their standardized and raw scales, i.e., the true treatment effect of performing IT or MICT alone.

We planned to conduct exploratory subgroup and moderation analyses across standardized effects for the following: work matched/unmatched, modality of training (ambulatory, cycling, or other), sex (proportion of sample as males), age (years), BMI (kg·m^2^), intervention characteristics including level of intensity of effort for IT (i.e., SIT vs HIIT), within-session IT interval number and duration and their interaction, duration of MICT sessions, the difference (i.e., MICT minus IT) in total weekly exercise duration (frequency × duration), and duration of interventions (weeks), method of body composition measurement (DXA, BIA, skinfolds, etc.), body composition region of measurement (upper, lower, trunk), and whether nutrition was controlled or uncontrolled. Note, we originally mentioned exploration of moderators for both standardised and unstandardised effects in our pre-registration. However, we ultimately opted to just explore standardised effects for absolute fat mass and FFM outcomes to compliment and explore heterogeneity in our main models. Furthermore, we adapted the operationalization of some moderators (e.g., intervention characteristics such as total weekly exercise duration) and some we could not explore fully given the number of effects available for certain sub-groups (these are noted in the analysis code). We also fit further (not pre-registered) models to examine adherence (number of attended sessions as a proportion of number of prescribed sessions) and dropout (number of participants dropped out as a proportion of number of participants randomized) proportions, as well as a Poisson regression model for adverse event count data (per 1000 person-sessions). All exploratory models utilized the same multilevel mixed-effects structure and specifications as the main models.

As a final exploratory (not pre-registered) analysis, we examined the variation in responses between both IT and MICT conditions. We sought to identify whether there was evidence of ‘true’ inter-individual variation from within-participant variability and/or participant-by-treatment interaction in responses to interventions by comparing the standard deviations for change scores with those of non-exercise control conditions [23,39]. We identified a mean-variance (on both the raw and log-transformed scales) relationship across studies for change scores (see https://osf.io/6zb8y/). Thus, we opted to adjust for this by employing a multilevel meta-regression of the log-transformed change score standard deviations, adjusted for the log-change score mean [40], calculated such that positive values showed that intervention condition (i.e., IT and MICT) variation exceeded control condition variation—thus suggesting evidence of ‘true’ inter-individual response variation. Where studies did not report change score standard deviations, or we were unable to calculate it directly, this was estimated using the imputed median pre-post correlation coefficient noted above as:$$SD_{change}=\sqrt{SD_{pre}^{2}+SD_{post}^{2}-\left(2\times r_{pre-post}\times SD_{pre}\times SD_{post}\right)}$$

Note that, given the different measurement devices used in individual studies, we accepted pragmatically the inherent assumptions built into this comparison of a constant Gaussian measurement error (i.e., that measurement error does not scale in a non-linear fashion with measured scores).

## 3. Results

### 3.1. Search Results

From the initially reviewed 2085 search results, a total of 56 studies were determined as meeting the inclusion criteria for our analysis. Two studies stated that body composition measures were performed, but did not report information on this outcome in the manuscript [41,42]. Attempts to obtain the data from the corresponding authors proved unsuccessful. Thus, we analyzed 54 studies that compared the effects of IT and MICT on measures of body composition. Table 1 presents a summary of the methods of the included studies. Table 2 presents descriptive information as to the included studies. Figure 2 shows the contour enhanced funnel plot for all effects from these studies. Inspection of the funnel plot did not reveal any obvious small study bias.

### 3.2. Methodological Quality

Study quality, as assessed by the PEDro scale, had a mean rating of 5.6, indicating that the overall pool of studies are of good quality. A total of 32 studies were rated as being of excellent quality, 21 studies were rated as being of good quality, and 1 study was rated as being of fair quality; no study in the analysis was deemed to be of poor quality. Individual scoring is available in the online Appendix A (https://osf.io/b28qd/).

### 3.3. Main Models

#### 3.3.1. Fat Mass

The main model for all fat mass effects (55 across 29 clusters (median = 1, range = 1–6 effects per cluster)) revealed a trivial standardized point estimate with a high precision for the interval estimate (−0.02 (95%CI = −0.07 to 0.04)), with moderate heterogeneity (*Q*_(54)_ = 79.08, *p* = 0.015, *I*^2^ = 36%). Figure 3 presents all standardized effects and interval estimates for fat mass outcomes across studies in an ordered caterpillar plot.

#### 3.3.2. Fat-Free Mass

The main model for all FFM effects (34 across 27 clusters (median = 1, range = 1–3 effects per cluster)) revealed a trivial standardized point estimate with a high precision for the interval estimate (−0.0004 (95%CI = −0.05 to 0.05)), with negligible heterogeneity (*Q*_(33)_ = 37.77, *p* = 0.26, *I*^2^ = 16%). Figure 4 presents all standardized effects and interval estimates for FFM outcomes across studies in an ordered caterpillar plot.

#### 3.3.3. GRADE Summary of Findings for Main Outcomes

For both fat mass and FFM there was a ‘high’ certainty of evidence with respect to the effects identified. It was deemed that there was no serious risk of bias, inconsistency, indirectness of evidence, or imprecision in estimates, nor were there other clear considerations impacting on certainty of evidence grading. The GRADE summary of findings table for our main outcomes is available in the Appendix A (https://osf.io/pcyvx/).

### 3.4. Secondary Analyses

Between condition treatment effect models on both the raw effect scales, and using relative fat outcomes (relative lean models not run due to limited data), showed similar outcomes to the main models reported. Thus, for brevity, these are presented in the Appendix A along with caterpillar plots (see folder “Outputs and Figures” > “Secondary Outcomes Outputs” > “Additional Between Condition Models” at https://osf.io/6karz/).

#### 3.4.1. Within-Condition Treatment Effects

All within-condition models are also available in the Appendix A (see folder “Outputs and Figures” > “Secondary Outcomes Outputs” > “Within Condition Models” at https://osf.io/6karz/) and here we report just the results for absolute fat and FFM outcomes on standardized and raw scales. In comparison to non-intervention control groups, the IT conditions resulted in small reductions in fat mass (Hedges’ g = −0.22 (95%CI = −0.36 to −0.08); kilograms = −0.20 (95%CI = −0.34 to −0.06)), and trivial increases in FFM (Hedges’ g = 0.13 (95%CI = 0.04 to 0.22); and kilograms = 0.11 (95%CI = −0.04 to 0.26)). The MICT conditions also produced small reductions in fat mass (Hedges’ g = −0.20 (95%CI = −0.36 to −0.04); kilograms = −0.25 (95%CI = −0.39 to −0.11)), and trivial increases in FFM (Hedges’ g = 0.07 (95%CI = −0.01 to 0.16); kilograms = 0.07 (95%CI = −0.02 to 0.15)).

#### 3.4.2. Sub-Group and Meta-Regression Analyses

Sub-group and meta-regression models were not run for absolute FFM standardized effects, given the negligible heterogeneity in the main model. When exploring sub-group and meta-regression models for absolute fat mass standardized effects, only two moderators—sex (proportion of males in sample; β = 0.0015 (95%CI = 0.00 to 0.0029)) and the number of intervals performed per training session by IT (β = −0.0032 (95%CI = −0.0052 to −0.0013))—appeared to have an effect, albeit this effect was relatively small for both covariates. Again, for brevity, all sub-group and meta-regression models are included in the Appendix A (see folder “Outputs and Figures” > “Secondary Outcomes Outputs” > “Sub-group and Meta-regression Models” at https://osf.io/6karz/).

#### 3.4.3. Adherence, Dropouts and Adverse Events

There was minimal difference in adherence or dropout proportions between conditions, which were relatively high and low, respectively. Adherence for IT was 89.1% (95%CI = 85.2% to 92.1%) and for MICT was 89.2% (95%CI = 84.5% to 92.7%), and dropouts for IT were 16.1% (95%CI = 11.4% to 22.2%) and for MICT were 20.1% (95%CI = 12.3% to 33.1%). Adverse events per 1000 person-sessions (i.e., the number of events per 1000 training sessions performed) were also relatively low with a minimal difference between conditions, with values of 1.15 (95%CI = 0.31 to 4.34) and 1.07 (95%CI = 0.51 to 2.24) for IT and MICT, respectively.

#### 3.4.4. Inter-Individual Response Variation

There was no clear evidence of ‘true’ inter-individual variation in responses for either IT or MICT conditions. The difference in intercepts when compared with CON conditions were −0.15 (95%CI = −0.35 to 0.05) and −0.02 (95%CI = −0.22 to 0.18) for IT and MICT, respectively (see figure in Appendix A: https://osf.io/3mazj/).

## 4. Discussion

This is the most comprehensive meta-analysis to date comparing IT and MICT on changes of measures of fat mass and FFM. Furthermore, GRADE assessment suggests high certainty in the evidence presented. Our findings provide novel insights into the use of different training strategies to bring about changes in body composition. Below, we discuss the results and practical implications of our data for each outcome.

### 4.1. Changes in Fat Mass

It has been speculated that IT may confer superior fat loss benefits compared to MICT, primarily mediated via a greater excess post-exercise oxygen consumption (EPOC) [97]. However, the overall magnitude of additional energy expenditure attributed to EPOC during IT is modest [98], and thus is unlikely to be of practical meaningfulness from a fat loss standpoint. Other proposed benefits of IT on fat reduction include enhancements in appetite suppression, fat oxidation, and circulating catecholamines and lipolytic hormones [98]. Despite this mechanistic rationale, our results do not support a superiority of IT on reductions in fat mass. Analysis of standardized between-group treatment effects showed similar changes for IT and MICT with both absolute fat mass as our primary outcome (Hedges’ g = (−0.02 (95%CI = −0.07 to 0.04)), and percentage body fat (Hedges’ g = −0.04 (95%CI = −0.08 to 0.01)). Raw absolute fat mass changes revealed a trivial point estimate of −0.17 kg favoring MICT, although the interval estimate ranged from −0.66 kg in favor of MICT to 0.31 kg in favor of IT. Comparison of raw relative (%) fat mass changes in fat mass revealed a small point estimate of −0.30% favoring MICT, but again, the interval estimate was imprecise, ranging from −0.63% in favor of MICT to 0.04% in favor of IT. Taken as a whole, these findings suggest that changes in fat loss are not meaningfully influenced by patterns of intensity of effort and duration (i.e., IT vs MICT) during exercise.

When compared to non-exercising controls, IT and MICT produced small reductions in fat mass, with minimal differences between conditions. The raw absolute fat loss amounted to −0.22 kg for IT and −0.25 kg MICT, with standardized Hedges’ g ES values of 0.22 and 0.20, respectively. Relative changes in fat mass for IT and MICT showed similarly small decreases vs controls, both on a raw (0.30% and 0.25%, respectively) and standardized (0.28 and 0.24, respectively) basis. None of the studies that included control conditions combined exercise with dietary intervention (i.e., caloric deficit) and thus, collectively, these data suggest that exercise alone induces a small magnitude of fat loss regardless of the patterns of intensity of effort and duration, at least under the methods employed in current research. More extreme volumes of exercise may be necessary to induce meaningful changes, irrespective of the intensity of effort. The observed changes in fat mass (~0.2 kg) in present studies and intervention examined are unlikely to be clinically or aesthetically meaningful in most populations. Indeed, these findings concur with earlier results from Keating et al. [8].

The lack of overall fat loss achieved in both IT and MICT can be attributed, at least in part, to the relatively low weekly exercise dose across studies (IT, median = 28 min duration (range = 3 min to 120 min); MICT, median = 120 min duration (range = 48 min to 250 min), and perhaps is confounded by a corresponding increase in energy intake [99] and/or reduction in non-exercise activity thermogenesis [100]. Tightly controlled research in identical twins shows that prolonged daily aerobic-type exercise can induce marked reductions in fat mass under conditions of constant energy and nutrient intake [101]. However, the time commitment needed to achieve these results (~100 min/day) is infeasible for the majority of the general public and is thus of limited practical relevance. Therefore, our findings underscore the importance of dietary prescription to facilitate weight loss; however, exercise may play an important supplementary role in the process [102].

In contrast to the recent meta-analysis from Sultana et al. [11], we did identify some moderate heterogeneity in our main model, leading us to explore possible moderators. For example, some evidence suggests that IT elicits greater reductions in abdominal adiposity compared to MICT [17]. Given the well-established association between android fat and cardiometabolic disease [103], such an outcome would potentially have major health implications if found to be true. However, our findings refute this contention, demonstrating similar changes in abdominal fat mass between conditions. Moreover, we found that relatively equal, albeit modest, fat loss occurred across the upper body, lower body and trunk regions regardless of condition, indicating that endurance-oriented exercise does not preferentially target specific fat deposits. Indeed, with the exception of sex and the number of intervals performed during IT training sessions, both of which also only had very trivial moderating effects, we did not identify any clear moderators of comparative treatment effects for fat mass.

### 4.2. Changes in Fat-Free Mass

Some researchers have proposed that the performance of aerobic exercise can elicit increases in skeletal muscle hypertrophy that are comparable to resistance exercise training [15]. However, a meta-analysis by Grgic et al. [104] refuted this contention, showing significantly greater hypertrophic adaptations from resistance training vs aerobic training, both at the whole-muscle and myofiber level. However, it should be noted that Grgic et al. [104] did not subanalyze the effects of endurance exercise intensity on hypertrophy outcomes. A recent review speculated that IT may provide sufficient stimulus to enhance muscle growth, particularly in middle-aged and older adults, as well as clinical populations [16]. Furthermore, some emerging evidence suggests that, although traditional resistance training and aerobic modality interventions may produce differing adaptations, when duration and intensity of effort are matched, similar strength and endurance adaptations may occur, although the impact on hypertrophy is less clear [105].

Our results suggest that endurance exercise intensity and duration may not mediate hypertrophic adaptations. Specifically, analysis of changes in FFM, both on an absolute and relative basis, demonstrated similar effects between IT and MICT. Between-condition treatment standardized effects for absolute changes in FFM were essentially zero ((−0.0004 (95%CI = −0.05 to 0.05)), and comparison of effects on the raw scale showed a small point estimate of 0.09 kg favoring IT, yet the interval estimate ranged from −0.18 kg in favor of MICT, to 0.35 kg in favor of IT. There were limited data reporting relative changes in FFM, with only three studies directly comparing MICT vs IT. Pooling of these data revealed a moderate magnitude of effect (−0.98%) favoring MICT. However, due to the lack of data, the confidence intervals around the point estimate were wide (−3.39% to 1.43%), and Hedges’ g values indicated a trivial standardized mean difference (0.17) with similarly wide interval estimates (−0.69 to 0.35). From a practical standpoint, these findings collectively suggest there may not be a meaningful difference between MICT and IT on absolute changes in FFM. 

Compared to non-exercising controls, our findings indicate trivial standardized effects for improvements in FFM for both conditions (IT, Hedges’ g = 0.13 (95%CI = 0.04 to 0.22); MICT, Hedges’ g = 0.07 (95%CI = −0.01 to 0.16)). IT showed absolute raw increases of 0.11 kg whereas MICT showed increases of 0.07, although both the lower bounds of the interval estimates included zero and the upper bounds did not reach particularly meaningful values. These data collectively suggest that neither MICT nor IT meaningfully affect FFM under the methods employed across studies, and call into question the claim that endurance-based exercise is a viable interventional strategy for promoting muscle hypertrophy. 

### 4.3. Exercise Adherence and Dropouts

Adherence was essentially identical between conditions, with both groups completing ~90% of sessions; dropouts were also similar and relatively low at ~13–17%. It has been argued that the intensity of effort of exercise influences core affective response [106], and that this is predictive of future intentions and behavior in relation to exercise [107]. However, a recent systematic review suggests that affective response may only differ trivially between IT and MICT, and that enjoyment responses may demonstrate a small effect in favor of IT [108]. Despite varying speculative theories regarding the intensity of effort during exercise, and its impact on affect or enjoyment, and subsequent behaviors, the results here suggest that adherence to IT and MICT is largely similar and relatively high, at least over the duration of the studies and under the conditions in which the interventions were employed. Indeed, it should be noted that exercise sessions in the included studies were carried out with the aid of programming from the respective research teams and were generally performed under direct supervision. It is well-established that programming and supervision have positive effects on exercise adherence [109]. Thus, our findings in this regard cannot necessarily be extrapolated to self-directed exercise programs. Given the high interindividual variability observed in the psychological response to endurance exercise [110], it would seem that allowing for a choice of training intensity would likely help to improve long-term adherence. Future research should endeavor to test this hypothesis under ecologically valid conditions. 

### 4.4. Adverse Events

Of the studies reporting adverse events, there was essentially no difference between IT and MICT. On the surface, this would seem to suggest that both conditions are similarly safe in the populations studied. However, most studies failed to report incidences of adverse events. Furthermore, some studies lacked clarity as to whether there was a comprehensive attempt to record all possible adverse events associated with the training intervention. Thus, data on the topic is somewhat limited, precluding the ability to draw strong inferences regarding the safety between protocols. 

A recent meta-analysis that examined the effects of supervised IT in patients with cardiovascular disease reported only five associated adverse cardiovascular events in approximately 17,000 training sessions: one major cardiovascular event, one minor cardiovascular, and three incidences of musculoskeletal issues. Although these findings appear to indicate that IT is generally safe, even in populations with non-communicable diseases and other health risks, results may be confounded by underreporting of adverse events in individual studies, and perhaps also by sampling bias for the types of individuals likely to participate in such studies. Researchers are thus encouraged to track and disclose the occurrence of such incidences in future studies on HIIT and MICT so that we can achieve a greater understanding of the risks associated with each strategy.

### 4.5. Inter-Individual Response Variation

Variance of treatment responses to IT and MICT has been relatively underexplored, despite numerous studies purporting that there may be inter-individual response variation to IT and MICT for a range of outcomes [19,20,21]. Indeed, some have argued that such variations may mask differences between IT and MICT for fat loss [22]. Evidence from the HERITAGE Family Study would genetically support this speculation, given that a putative dominant locus accounting for 31% of variance in fat mass changes was found [111]. However, we found no evidence of ‘true’ inter-individual variability in responses to either IT or MICT. This is in agreement with findings from a recent meta-analysis of aerobic exercise in overweight individuals and children and adolescents with obesity on fat loss [18]. Given our findings, and the relatively low heterogeneity across the main models for outcomes, the majority of apparent differences in study level results and apparent ‘response heterogeneity’ are likely attributable to sampling variance and random within-subject variability.

### 4.6. Limitations

The present meta-analysis has several limitations that must be taken into account when attempting to draw practical inferences on the effects of IT vs MICT on measures of body composition. First and foremost, only three studies prescribed dietary energy restrictions for the interventional protocol. Thus, it is not clear whether one exercise strategy may be superior to another when combined with a nutritional intervention. Second, only one study supplemented the exercise intervention with a resistance training component. It is possible that differences in intensity and duration between IT and MICT protocols might alter responses when combined with resistance training. Although recent evidence questions whether there is an interference effect from concurrent training, at least for hypertrophy [112], the specific roles of endurance exercise intensity and duration upon fat mass under these conditions have yet to be elucidated. Third, very few studies involved trained athletes, and the vast majority of subjects would be considered to be overweight/obese. Thus, it remains to be determined how differences in endurance exercise intensity and duration may affect body composition outcomes in lean and athletic populations. Moreover, the majority of included studies examined outcomes in younger to middle-aged adults, limiting our ability to draw conclusions about the effects of IT and MICT on older populations. Fourth, although we were able to separate studies that had included control groups for the purpose of a ‘within-condition’ analysis of the true treatment effects for IT and MICT, in addition to exploration of interindividual response variability, these were secondary exploratory analyses. Our search strategy and inclusion were not optimized to identify all studies that included either IT or MICT and a non-training CON condition. However, our estimates for within-group IT effects were not dissimilar to those reported by Sultana et al. [11] for IT vs CON, who did include studies with either an MICT or a non-training CON condition. Finally, our analysis is specific to body composition changes and does not take into account the other potential effects of the different interventional exercise strategies. Some evidence indicates that higher intensities of exercise may confer superior health-related benefits such as improvements in glucose control, blood pressure, vascular function, and cardiorespiratory fitness [113]. Thus, the use of a given endurance exercise strategy should consider individual goals in combination with abilities and preferences.

## 5. Conclusions

Our findings provide compelling evidence that the patterns of intensity of effort and duration during endurance exercise has minimal influence on longitudinal changes in fat mass and FFM. From a practical standpoint, this implies that individuals can choose the intensity of effort and duration combination (i.e., IT or MICT) that best suits their needs and lifestyle. As a general rule, there is an efficiency/effort tradeoff along the intensity of effort spectrum, whereby IT requires less time but more effort than MICT to promote alterations in body composition. Given that exercise adherence is of paramount concern, personal preference should thus guide prescription.

Our findings also indicate that structured exercise only has minor effects on fat loss regardless of intensity of effort and duration when performed at relatively modest doses; the amount of exercise required to achieve practically meaningful changes in this outcome seems to be unrealistic for most individuals. It is much easier to create an energy deficit from dietary restriction, which, therefore, should be the focus of weight loss interventions. However, exercise may help to preserve FFM and functional performance during periods of energy restriction [114], as well as facilitate sustenance of weight loss in combination with a dietary intervention [115]. Thus, it should be considered an important adjunct to nutritional approaches for those who endeavor to alter their body composition.

## Figures and Tables

**Figure 1 sports-09-00155-f001:**
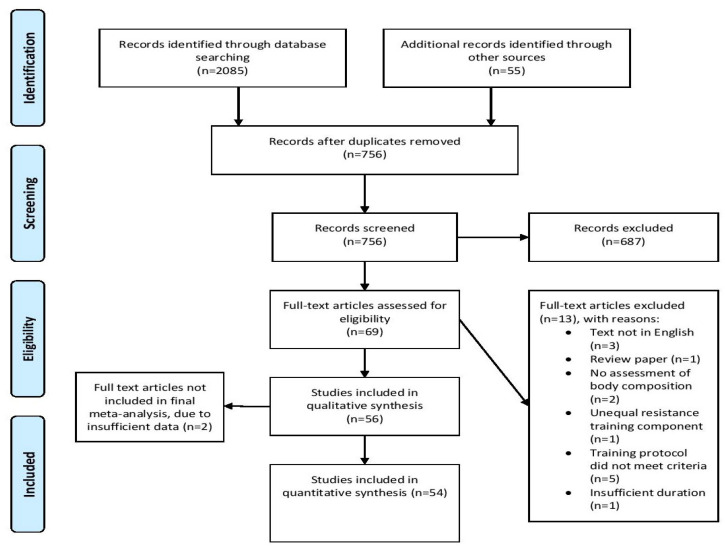
Flow chart of PRISMA search.

**Figure 2 sports-09-00155-f002:**
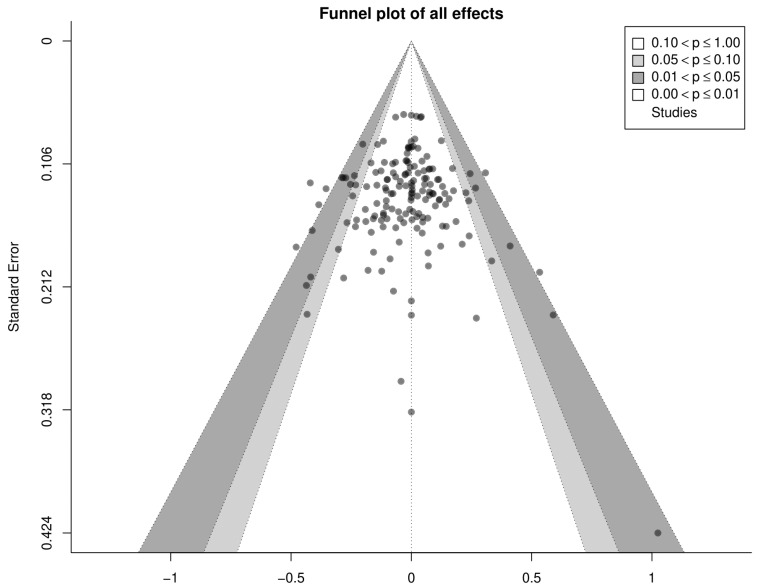
Contour enhanced funnel plot for all effects from the included studies.

**Figure 3 sports-09-00155-f003:**
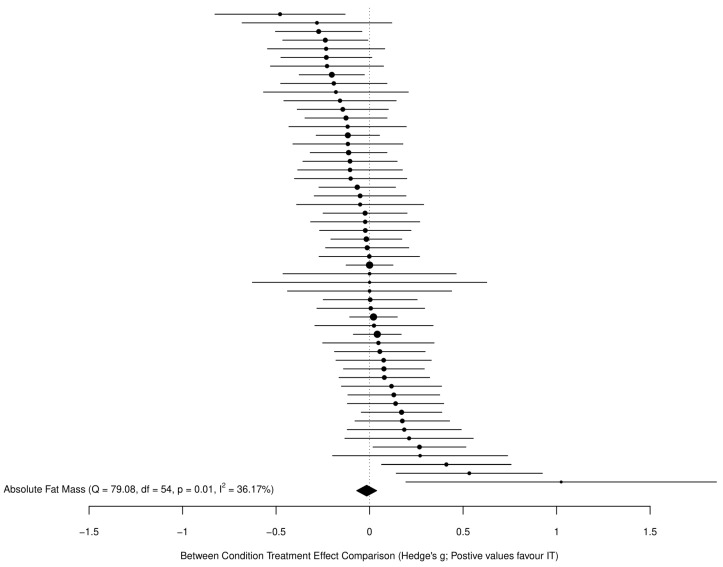
Standardized effects and interval estimates for fat mass outcomes across all studies in an ordered caterpillar plot.

**Figure 4 sports-09-00155-f004:**
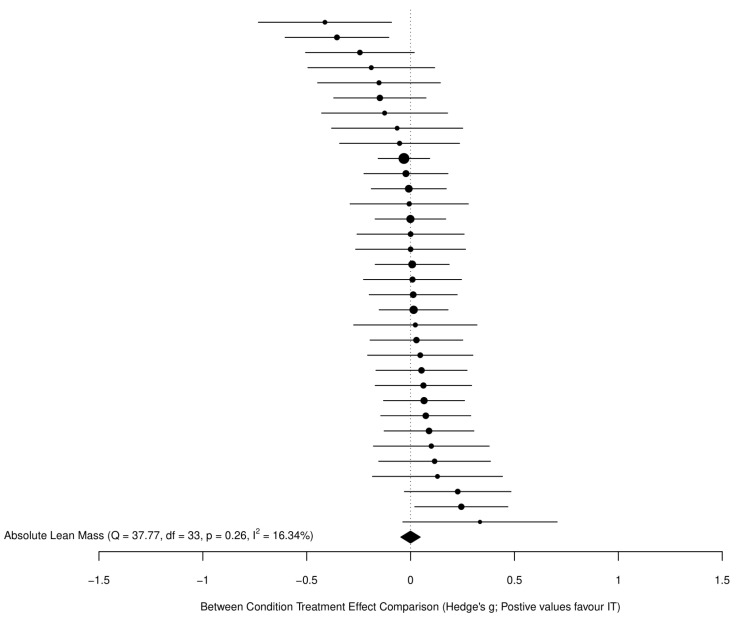
Standardized effects and interval estimates for FFM outcomes across studies in an ordered caterpillar plot.

**Table 1 sports-09-00155-t001:** Methods of included studies.

Study	Sample Population (age)	Duration (weeks)	Group (n)	Modality/Intensity	Frequency (weeks)	Time per Session	Body Composition Method
[43]	Obese children (aged 8–12 years)	12	MICT: 15 IT: 15	MICT: 80% of peak heart rate IT: 3–6 sets of 60 s sprint at 100% of the peak velocity with 3 min active recovery period at 50% of the exercise velocity.	MICT: 2× IT: 2×	MICT: 30–60 min IT: 9–18 min	BIA
[44]	Young adults with intellectual disabilities (aged 16–18 years)	15	MICT: 15 IT: 17 CON: 14	MICT: Cycling, walking/running, stepping at 30% peak watts IT: First 7 weeks: 10, 15 s sprint bouts at ventilatory threshold (100 + RPM), 45 s recovery period at 50 RPM Weeks 8–15: intensity increased to 110% VT	MICT: 2× IT: 2×	MICT: 40 min IT: 40 min	BIA
[45]	Down syndrome adults (mean age 34 years)	12	MICT: 13 IT: 13 CON: 16	MICT: Continuous cycling/walking at 70–80% VO_2_ peak, 85% after week 6 IT: 10 × 30 s sprints, 90 s rest period	MICT: 3× IT: 3×	MICT: 30 min IT: 30 min	BIA
[46]	Adolescents (mean age 16 years)	7	MICT: 16 IT: 17 CON: 24	MICT: Running at moderate intensity at 70% VO_2_ max (VO_2_ max retested at week 4) IT: 4–6 all-out sprints × 30 s, 30 s recovery period (20 s recovery period for week 7)	MICT: 3× IT: 3×	MICT: 20 min IT: 16 min	Skinfolds
[47]	Children (mean age: 11 years)	8	MICT: 16 IT: 18	MICT: Running at 65–75% HRmax IT: 3–6 bouts, 20 s max-effort sprint, 60 s rest period (40 s at week 5, 20 s at week 8)	MICT: 3× IT: 3×	MICT: 4–9 min IT: 4–9 min	BIA
[48]	Obese adults (mean age: 39 years)	12	MICT: 6 IT: 6	MICT: Brisk walking at 4 METs IT: 4–7 intervals at a 2:1 ratio, then 5 boxing drills × 3 intervals at a 2:1 ratio, RPE 15–17 (>75% HRmax)	MICT: 4× IT: 4×	MICT: 50 min IT: 50 min	Skinfolds
[49]	Obese adult men (mean age: 25 years)	4	MICT: 8 IT: 8	MICT: Continuous cycling @ 65% VO_2_ peak IT: 4–7 sprints × 30 s at 200% W-max, 120 s at 30 W in-between	MICT: 5× IT: 3×	MICT: 40–60 min IT: 10–17.5 min	DXA
[50]	Colorectal cancer survivors (mean age: 62 years)	4	MICT: 14 IT: 21	MICT: Continuous cycling at 50–70% HRpeak IT: 4 cycling intervals × 240 s at 85–95% HRpeak, 180 s active rest	MICT: 3× IT: 3×	MICT: 50 min IT: 38 min	DXA
[51]	Obese children (aged 7–16 years)	12	MICT: 22 IT: 18 CON: 16	MICT: 60–70% HRmax IT: 4 intervals of 240 s at 85–95% HRmax, 180 s active recovery at 50–70% HRmax	MICT: 3× IT: 3×	MICT: 44 min IT: 28 min	DXA and ADP
[52]	Adult men at risk for insulin resistance (mean age: 48 years)	12	MICT: 16 IT: 21	MICT: Worked towards 6 kcal/kg per week for 6 weeks (+2 per week until 12 kcal/kg per week) treadmill at 50–70% VO_2_ max IT: Performed MICT protocol until week 6 then transitioned, 2–8 bouts of 60 s at 90–95% VO_2_ max, 60 s recovery period at 50% VO_2_ max	MICT: 3–4× IT: 3–4×	MICT: Work-dependent IT: 4–16 min	DXA
[53]	Obese young women (mean age: 22 years)	8	MICT: 7 IT: 7 CON: 6	MICT: Continuous treadmill at 50–70% at HRpeak IT: 4 intervals of 240 s at 85–95% HRpeak, 180 s active rest	MICT: 3× IT: 3×	MICT: 41 min IT: 33 min	BIA
[54]	Sedentary young men (mean age: 21 years)	8	MICT: 6 IT: 6	MICT: Continuous treadmill at 70–80% VO_2_ max IT: 12 intervals of 60 s at 90–110% VO_2_ max, 60 s rest period	MICT: 3× IT: 3×	MICT: 30 min IT: 30 min	DXA
[55]	Obese young men (aged 17–22 years)	6	MICT: 13 IT: 15	MICT: Cycling at 55–65% VO_2_ peak IT: 4 intervals of 240 s at 15% APmax, then 30 s at 85% APmax, then 120 s at 15% APmax	MICT: 5× IT: 3×	MICT: 45–60 min IT: 20 min	DXA
[56]	Overweight men (aged 20–40 years)	12	MICT: 12 IT: 10	MICT: Walking/jogging at 65% HRmax IT: 6–12 intervals of 60 s at 90–95% HRmax, 60 s active rest	MICT: 3× IT: 3×	MICT: 18–35 min IT: 12–24 min	DXA
[57]	Sedentary young men (mean age: 27 years)	12	MICT: 10 IT: 9 CON: 6	MICT: Continuous cycling at 70% HRmax IT: 3 all-out intervals of 20 s at 0.5 kg/kg resistance, 120 s low-intensity active rest	MICT: 3× IT: 3×	MICT: 45 min IT: 10 min	ADP
[58]	Male police officers (mean age: 39 years)	8	MICT: 11 IT: 11	MICT: Continuous running at 60–75% V-shuttle max IT: 7–10 intervals of 85–100% V-shuttle max (V-shuttle based on individual shuttle test results)	MICT: 3× IT: 3×	MICT: 27.8-33.4 min IT: 14.8–19.1 min	DXA
[59]	Overweight young women (mean age: 20 years)	6	MICT: 29 IT: 23	MICT: Continuous cycling at 60–70% HRR IT: 5–7 all-out intervals of 30 s, 240 s active recovery	MICT: 3× IT: 3×	MICT: 20–30 min IT: 22.5–31.5 min	DXA
[60]	Healthy, sedentary older adults (aged 55–79 years)	8	MICT: 14 IT: 15 CONT: 14	MICT: 70% of peak heart rate IT: 4 × 4 min intervals at 90% of peak heart rate with 3 × 3 min active recovery periods at 70% of peak heart rate.	MICT: 4× IT: 4×	MICT: 47 min IT: 40 min	DXA
[61]	Inactive, overweight adults (aged 18–55 years)	12	MICT: 11 IT: 11 CONT: 11	MICT: 50–65% VO_2_peak IT: cycling, 4–6 sets of 30–60 s at 120% VO_2_peak with 120–180 s at 30 W.	MICT: 3× IT: 3×	MICT: 30–45 min IT: 20–24 min	DXA
[62]	Obese adolescents (mean age: 13 years)	12	MICT: 15 IT: 14	MICT: 60–70% of VO_2_max IT: running for 2 min at 80–90% of VO_2_max followed by recovery periods of 1 min.	MICT: 3× IT: 3×	MICT: 30–40 min	Skinfolds
[63]	Overweight, inactive adults (aged 35–60 years)	12	MICT: 17 IT (AIT): 11 IT (MVIT): 16	MICT: Walking, 65–75% of HRmax IT (AIT): jogging, 4 cycles of 4 min at 85–95% HRmax followed by 3 min recovery at 65–75% HRmax. IT (MVIT): 30 s of “all out” exercise followed by 4 min of low intensity recovery.	MICT: 3× IT (AIT): 3× IT (MVIT): 3×	MICT: 48 min IT (AIT): 40 min IT (MVIT): 24.5–40 min	BIA
[64]	Healthy, recreationally active young adults (mean age: 23 years)	6	MICT: 10 IT: 10	MICT: running, 65% of VO_2_max IT: 4–6 bouts of 30 s maximal running efforts with 4 min of recovery (active recovery encouraged)	MICT: 3× IT: 3×	MICT: 30–60 min IT: 18–27 min	ADP
[65]	Overweight, untrained men (aged 28–46 years)	10	MICT: 7 IT: 7	MICT: 50% of VO_2_max IT: 25 sets of 80 s at 35% VO_2_max followed by 40 s at 80% VO_2_max.	MICT: 3× IT: 3×	MICT: 50 min IT: 50 min	Skinfolds
[66]	Adults with type 2 diabetes (mean age: 59 years)	52	MICT: 24 IT: 19 CONT: 24	MICT: cycling 40–60% of HRR IT: cycling, 2 min at 70–80% of HRR with 1 min at 40–60% of HRR. 1 min at 90% of HRR with 1 min resting at 40–60% of HRR.	MICT: 3× IT: 3×	MICT: 45 ± 7.1 min IT: 33.1 ± 6.4 min	DXA
[67]	Postmenopausal women with type 2 diabetes (mean age: 69 years)	16	MICT: 8 IT: 8	MICT: 55–60% of individual HR reserve IT: 60 cycle (maximum) of 8 s at 77–85% HRmax with active recovery of 20–30 rpm for 12 s.	MICT: 2× IT: 2×	MICT: 40 min IT: 25 min	DXA
[68]	Children (aged 7–9 years)	12	MICT: 56 IT: 38	MICT: 20 min of moderate-intensity aerobic exercises and games followed by 20 min of sport. IT: 20 min of 10–20 s of high-intensity intermittent exercises followed by 20 min of sports activities.	MICT: 2× IT: 2×	MICT: 40 min IT: 40 min	BIA
[69]	Sedentary Obese Adults (aged 34 years)	12	MICT: 14 IT: 16 ½ IT: 16	MICT: 70% of peak HR IT: 8 s of maximal intensity sprint intervals on a bike at 85–90% of peak HR, with 12 s rest intervals pedaling as slow as possible. Sequence continued until the 250 kcal target was met. ½ IT: Same as IT but with a 125 kcal target.	MICT: 3× IT: 3× ½ IT: 3×	MICT: 32 min(avg.) IT: 20 min(avg.) ½ IT: 10 min (avg.)	DXA
[70]	Sedentary adult men (aged 29 years)	8	MICT: 12 IT: 12	MICT: 60-65% VO_2_max IT: 3, 3 min intervals of high intensity cycling at 80–85% VO_2_max with 2 active rest intervals.	MICT: 3× IT: 3×	MICT: 45 min IT: 18 min total including 5 min of combined warm-up and cool down.	DXA
[71]	Adult men with metabolic syndrome (mean age: 48 years)	8	MICT: 13 IT: 13	MICT: cycling at 60–65% of VO_2_peak IT: 3 sets of 3 min cycling at 80–85% VO_2_peak with a 2 min active rest between sets at 50% VO_2_peak	MICT: 3× IT: 3×	MICT: 45 min IT: 18 min	DXA
[72]	Sedentary premenopausal women (mean age: 45 years)	15	MICT: 21 IT: 21 CON: 20	MICT: Moderate intensity swimming at ~70% HRmax. IT: 6–10 × 30 s all-out swimming with 2 min recovery in between each bout at.~90% HRmax	MICT:3× IT: 3×	MICT: 1 h IT: 15–25 min total.	DXA
[73]	Overweight adults (mean age: 40 years)	12	MICT: 8 IT: 8 CON: 7	MICT: Biking at 10% lower than anaerobic threshold. IT: Biking at 20% above anaerobic threshold with an exercise:pause ratio of 2:1.	MICT: 3× IT: 3×	Both groups completed 20 min in the first week, with increments of 10 min per week until a total of 60 min per session was reached in the fourth week.	BIA
[74]	Obese adolescents (mean age: 15 years)	12	MICT: 13 IT: 16	MICT: Boxing and Nordic walking at 60–75% of maximal HR. IT: 4 to 6 intervals of 2 min–2 min 30 s in duration at 90–95% of HRmax interspersed by 1 min 30 s intervals at 55% of HRmax	MICT: 3× IT: 3×	MICT: 40 to 60 min. IT: 24 to 32 mins	DXA
[75]	Obese adolescents (mean age: 14 years)	4	MICT: 8 IT: 10	MICT: 65% HRmax IT: 1 min vigorous treadmill exercise at 80% to 90% HRmax interspersed with 2 min recovery intervals at 60% HRmax	MICT: 3× IT: 3×	MICT: 50 min	BIA
[76]	Recreationally active men (mean age: 21.7 years)	7	MICT: 7 IT: 8	MICT: Cycling at 60% of VO_2_max. IT: 4–6 Wingate sprints (resistance = 7.5% of subject BW) with 4.5 min recovery	MICT: 3× IT: 3×	MICT: 30–50 min IT: 30 min	Skinfolds
[77]	Untrained men (mean age: 33 years)	12	MICT: 9 IT; 8 CON:11	MICT: 80% HRmax IT: Five intervals of 2 min of near-maximal running (HR above 95% of their HRmax at the end of the 2 min period interspersed by 1 min rest.	MICT: 3× IT: 2× (attempted 3 but accomplished 2 on average due to injuries or other reasons)	MICT: 1 h IT: 20 min	DXA
[78]	Sedentary obese males (mean age: 48.4 years)	12	MICT: 13 IT: 20	MICT: Cycling at 60–65% VO_2_max IT: 3 sets of 180 s cycling at 80–85% VO_2_max with 120 s recovery period at 50% VO_2_max	MICT: 3× IT: 3×	MICT: 40 min IT 13 min	DXA
[79]	Untrained women (mean age: 28.4 years)	6	MICT: 12 IT: 11	MICT: Cycling at 70% HRmax IT: Cycling 15 sets 60 s at 90% HRmax with 30 s recovery period at 60% HRmax	MICT: 3× IT: 3×	MICT: 29 min IT: 22 min	Skinfolds
[80]	Untrained obese women (mean age: 46 years)	12	MICT: 12 IT: 18	MICT: Deep water running at 65–85% HRR IT: Deep water running 8–15, 15 s sprints with 30s recovery interspersed with 5–14 min intervals at 70–75% HRmax	MICT: 3× IT: 3×	MICT: 47 min IT: 47 min (including recovery periods)	Skinfolds
[81]	Healthy physically inactive adults (mean age: 32 years)	12	MICT: 9 IT: 11	MICT: Treadmill, 60–80% HRR IT: Treadmill, 4 sets 240 s at 85–95% peak HRR with 240 s recovery period at 65% peak HRR	MICT: 3× IT: 3×	MICT: 20–65 min (including warm up and cool down) IT: 35 to 55 min (including warm up and cool down)	BIA
[82]	Adults with metabolic syndrome (mean age: 57 years)	16	MICT: 21 IT (a): 22 IT (b): 23	MICT: Cycling, 60–70% of peak heart rate IT (a) Cycling, 4HIIT group-4 240 s sets at 85–95% peak heart rate with 180 s recovery period at 50–70% peak heart rate IT (b) Cycling, 1HIIT group- 1 set 240 s at 85–95% peak heart rate with 180 s cool down at 60–70% peak heart rate	MICT: 5× IT (a): 3× IT (b): 3×	MICT: 30 min IT (a): 4HIIT, 38 min (including warm up and cool down). IT (b): 1 HIIT, 17 min (including warm up and cool down)	DXA
[83]	Sedentary adults (mean age: 31 years)	8	MICT: 7 IT (a): 9 IT (b): 11	MICT: Cycling, 65–75% HRmax IT (a): 2 × 4 HIIT, cycling 2 sets 240 s at 85–95% HRMax with 120 s active rest IT (b): 5 × 1 HIIT, cycling 5 sets 60 s at 85–95% HRMax with 60 s active rest	MICT: 2× IT (a): 2× IT (b): 2×	MICT: 38 min (including warm up and cool down) IT (a): 15 min (including warm up and cool down) IT (b): 14 min (Including warm up and cool down)	BIA
[84]	Sedentary males (age not reported)	4	MICT:12 IT: 12	MICT: Cycling, 45% VO_2_max IT: Cycling, 10 sets, 60 s at 85% VO_2_max with 30 s rest period between sets	MICT:3× IT:3×	MICT: 22 min IT: 15 min (including rest periods)	BIA
[85]	Obese adults (mean age: 46 years)	12	MICT: 13 IT:14	MICT: Treadmill, 60–70% HRMax IT: Treadmill, 4 sets 240 s at 85–95% HRMax with 180 s rest periods at 50–60% HRMax	MICT:3× IT: 3×	MICT: 47 min IT: 42 min (including warm up and cool down)	DXA
[86]	Sedentary males (mean age: 2 years)	6	MICT: 8 IT: 8	MICT: Cycling, ~65% VO_2_Peak IT: Cycling, four to six, 30 s ‘all out’ sprints (Wingate test) with 270 s rest between each test	MICT:5× IT:3×	MICT: 40 to 60 min IT: 20–30 min (including rest periods)	DXA
[87]	Overweight adults (mean age: 42 years)	10	MICT: 44 IT: 46	MICT: Cycling, ~70% MHR IT: Cycling, >90% MHR, repeated sprints of 15–60 s, interspersed with periods of recovery cycling	MICT: 5× IT: 3×	MICT: 30–45 min IT: 18–25 min	BIA
[88]	Trained young adults (mean age: 19 years)	8	MICT: 7 IT: 7	MICT: Rowing, Blood Lactate Concentrations of 2–3 mmol/L IT: Rowing, eight, 2.5 min intervals at 90% of mean 4 min maximal power output achieved during the incremental exercise test. Recovery duration was until HR returned 70% MHR, at 40% of mean maximal power output	MICT: 2× IT: 2×	MICT: 35/40 min IT: 27–55 min (including recovery)	DXA
[89]	Overweight Young Adults (mean age: 20 years)	12	MICT: 16 IT: 17 CON: 19	MICT: Walking/Jogging, HR associated with 50% of VO_2_max IT: Running, five, 3 min intervals at the HR associated with 85% VO_2_max with 3 min active rest at HR associated 50% VO_2_max between each interval	MICT: 5× IT: 5×	MICT: 55 min IT: 42 min (Including warm up and cool down)	DXA
[90]	Overweight Males (mean age: 31 yrs)	12	MICT: 10 IT: 10 CON: 10	MICT: Cycling, ~60% VO_2_peak IT: Cycling, 15 s at a power output equivalent to ~170% VO_2_peak with an active recovery period of 60 s at a power output equivalent to ~32% VO_2_peak Relative total work was matched between both groups	MICT: 3× IT: 3×	MICT: 30–45 min IT: 30–45 min	DXA
[91]	Obese Children (mean age: 15 years)	6	MICT: 13 IT: 14	MICT: Cycling, 65–70% APMHR IT: Cycling, ten, 2 min bouts at 90–95% APMHR, with 1 min of active recovery at 55% APMHR between each bout	MICT: 3× IT: 3×	MICT: 40 min IT: 40 min (Including warmup and cool down)	ADP
[92]	Healthy Untrained Adults (aged 18–32 years)	12	MICT A: 14 MICT B: 18 IT: 15	MICT A: Running, 4 m, 75% MHR MICT B: Running, 2 m, 75% MHR IT: Running, 8 bouts of 60 s intervals at 90% MHR followed by 180 s rest between each bout	MICT: 3× IT: 3×	MICT A: ~32 min/~500 cal/session MICT B: ~16 min/~250 cal/session IT: 29 min	Hydrostatic densitometry
[93]	Healthy Inactive Young Females (mean age: 21 years)	15	MICT: 15 IT: 15	MICT: Cycling, 60% VO_2_peak IT: Cycling, maximum of 60 bouts of 8 s:12 s ratio of sprinting and slow pedaling	MICT: 3× IT: 3×	MICT: 20–50 min IT: 15–30 min (Including warmup and cool down	DXA
[94]	Obese Adults (mean age: 43 years)	8	MICT: 6 IT: 7 CON: 8	MICT: Cycling, 50–65% VO_2_ peak IT: 2 min ratio of high to low intensity of 90–105% VO_2_peak and 30–45% VO_2_peak	MICT: 4× IT: 4×	MICT: 30 min IT:30 min (Including recovery)	DXA
[95]	Overweight Adults (mean age: 56 years)	11	MICT: 12 IT: 13 CON: 7	MICT: Cycling, 50% Wpeak IT: Cycling, 1 min at 95% Wpeak, with 1 min active recovery at 20% Wpeak between each bout.	MICT: 3× IT: 3×	MICT: 135 min IT: 75 min (Including the warmup)	DXA
[96]	Overweight Young Females (mean age: 21 years)	12	MICT: 15 IT: 15 CON: 13	MICT: Cycling, 60% VO_2_max until 300 kJ of work is reached IT: Cycling, repeated 4 min bouts at 90% VO_2_max with 3 min passive recovery between bouts until 300 kJ of work is reached	MICT: 3–4× IT: 3–4×	MICT: Until 300 kJ of work was reached IT: Until 300 kJ of work was reached	DXA

Abbreviations: MICT = moderate intensity continuous training; IT = interval training; CON = control; BIA = bioelectrical impedance analysis; DXA = dual energy x-ray absorptiometry; ADP = air displacement plethysmography.

**Table 2 sports-09-00155-t002:** Descriptive characteristics.

Characteristic	Number of Groups within Studies = 60
Age (years)	30 (21, 44)
Unknown	1
Sex (% Male)	54 (32, 100)
BMI (kg·m^2^)	28.3 (25.4, 30.5)
Unknown	7
Training Status	
Recreationally Active	1 (1.7%)
Trained	1 (1.7%)
Untrained	58 (97%)
Was Nutrition Controlled?	
No	31 (52%)
Yes	29 (48%)
Included Caloric Deficit?	
No	57 (95%)
Yes	3 (5.0%)
Include Resistance Training Intervention?	
No	59 (98%)
Yes	1 (1.7%)
Were IT/MICT Work-Matched?	
No	34 (57%)
Yes	25 (42%)
Yes, matched for time	1 (1.7%)
Intervention Duration (weeks)	12 (8, 12)
IT Frequency (median days per week)	
2	8 (13%)
3	44 (73%)
3.5	2 (3.3%)
4	3 (5.0%)
4.5	2 (3.3%)
5	1 (1.7%)
MICT Frequency (median days per week)	
2	8 (13%)
3	38 (63%)
3.5	2 (3.3%)
4	3 (5.0%)
4.5	2 (3.3%)
5	7 (12%)
Was IT Performed as SIT or HIIT?	
HIIT	45 (75%)
SIT	15 (25%)
IT Interval Number Performed	5 (4, 10)
Unknown	5
IT Interval Duration (median s)	60 (30, 180)
IT Total Exercise Duration (min)	9.4 (3.4, 16.0)
MICT Session Duration (min)	38 (30, 45)
Unknown	3
IT Adherence (% Sessions)	90 (83, 98)
Unknown	24
MICT Adherence (% Sessions)	90 (84, 97)
Unknown	25
IT Adverse Event Number	
0	12 (63%)
1	2 (11%)
2	2 (11%)
3	1 (5.3%)
4	1 (5.3%)
5	1 (5.3%)
Unknown	41
MICT Adverse Event Number	
0	10 (67%)
1	3 (20%)
2	2 (13%)
Unknown	45

Note: Values are Median (IQR) for continuous variables, and n (%) for categorical.

## Data Availability

All data, code, materials and supplementary analyses are openly available on the Open Science Framework page at: https://osf.io/6karz/.

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
