# Peer review of "Slow and Steady, or Hard and Fast? A Systematic Review and Meta-Analysis of Studies Comparing Body Composition Changes between Interval Training and Moderate Intensity Continuous Training"

_sports, 2021, doi:10.3390/sports9110155_

Round 1

Reviewer 1 Report

I would like to express my compliments to the authors for this huge  work, I just have a few suggestions.

To better understand the changhes observed in body composition I would add a table showing the percentage of  methods used to detect it. For examble: in how many of the studied works was used BIA, or skifold etc.

As the studies included in the review are heterogeneous I would have expressed the variations in body composition in percentage.  I also suggest to use Fat Free Mass (FFM) instead of lean mass.

Author Response

I would like to express my compliments to the authors for this huge  work, I just have a few suggestions.
AUTHOR RESPONSE: Thank you very much for reviewing our manuscript. We appreciated the positive feedback, and have addressed your comments on a point-by-point basis below.

To better understand the changes observed in body composition I would add a table showing the percentage of  methods used to detect it. For examble: in how many of the studied works was used BIA, or skinfold etc.
AUTHOR RESPONSE: Good suggestion, we have added a column to Table 1 that lists the body composition measurement method for each study.

As the studies included in the review are heterogeneous I would have expressed the variations in body composition in percentage.  I also suggest to use Fat Free Mass (FFM) instead of lean mass.
AUTHOR RESPONSE: As per your suggestion, we have changed “lean mass” to FFM throughout the manuscript

Reviewer 2 Report

Excellent work. Just Minor corrections should be done, as well as:

 - P7, R 264: ...trunk),  and....... - extra tab which should be deleted. 

 - P19, R 420 - 429: Font number is smaller than in the rest of the text. Please, do proper corrections.

 - P19, R 432 - 433: (IT, median = 28 mins [range = 3 mins to 120 mins]; MICT, median = 120 mins [range = 48 to 250]) - Here it is necessary to emphasize exercise training period i.e. duration (on average and overall range of time) those body mass changes are.

 - P21, R 523: Diseased population - is when someone is not healthy, i.e. when he is sick or infectiously ill. It is better to change this into: even in a population with non-communicable diseases and other health risks.....

  • P22, R 598 - ...: For some references, the title of the reference is written with a capital letter for each word (4, 6, 11 ......), and for some with a lower case letter (1, 2, 3, 5,....). It is necessary to unify it in accordance with the technical requirement.

Anyway, congratulations.

Best.

Author Response

Excellent work. Just Minor corrections should be done, as well as:

AUTHOR RESPONSE: Thank you very much for reviewing our paper, and for the positive feedback. We have addressed your comments on a point-by-point basis, hopefully to your satisfaction.

 - P7, R 264: ...trunk),  and....... - extra tab which should be deleted. 

AUTHOR RESPONSE: Good catch. We revised accordingly.

 - P19, R 420 - 429: Font number is smaller than in the rest of the text. Please, do proper corrections.

AUTHOR RESPONSE: We have revised so that the fonts are consistent.

 - P19, R 432 - 433: (IT, median = 28 mins [range = 3 mins to 120 mins]; MICT, median = 120 mins [range = 48 to 250]) - Here it is necessary to emphasize exercise training period i.e. duration (on average and overall range of time) those body mass changes are.

AUTHOR RESPONSE: We have revised to indicate this refers to duration

 - P21, R 523: Diseased population - is when someone is not healthy, i.e. when he is sick or infectiously ill. It is better to change this into: even in a population with non-communicable diseases and other health risks.....

AUTHOR RESPONSE: We have revised as per your suggestion.

- P22, R 598 - ...: For some references, the title of the reference is written with a capital letter for each word (4, 6, 11 ......), and for some with a lower case letter (1, 2, 3, 5,....). It is necessary to unify it in accordance with the technical requirement.

AUTHOR RESPONSE: Thanks for noting this issue. We have gone through the bibliography and revised so that the cases are consistent across citations.
